# Nomilin from Yuzu Seed Has In Vitro Antioxidant Activity and Downregulates Melanogenesis in B16F10 Melanoma Cells through the PKA/CREB Signaling Pathway

**DOI:** 10.3390/antiox11091636

**Published:** 2022-08-23

**Authors:** Moon-Hee Choi, Seung-Hwa Yang, Nam Doo Kim, Hyun-Jae Shin

**Affiliations:** 1Department of Beauty and Cosmetology, Graduate School of Industrial Technology and Entrepreneurship, Chosun University, Gwangju 61452, Korea; 2Department of Chemical Engineering, Graduate School of Chosun University, Gwangju 61452, Korea; 3VORONOI BIO Inc., Incheon 21984, Korea

**Keywords:** yuzu tree, seed husk, nomilin, antioxidant, antimelanogenic, tyrosinase inhibitor, skin-whitening agent, reactive oxygen species

## Abstract

Yuzu (*Citrus junos*) is a citrus plant native to Asian countries, including Korea, Japan, and China. Yuzu peel and seed contain abundant vitamin C, citric acid, and polyphenols. Although the antioxidative and antimelanogenic activities of other citrus fruits and yuzu extract have been reported, the tyrosinase inhibitory activity of the limonoid aglycone contained in yuzu seed extract is unknown. We separated yuzu seeds into the husk, shell, and meal and evaluated antioxidant activity of each. The limonoid glucoside fraction of the husk identified nomilin, a novel tyrosinase inhibitor. We performed tyrosinase inhibitory activity and noncompetitive inhibition assays and docking studies to determine nomilin binding sites. Furthermore, we evaluated the antioxidative mechanism and antimelanogenic activity of nomilin in B16F10 melanoma cells. The concentration of nomilin that did not show toxicity was <100 µg/mL. Nomilin suppressed protein expression of TYR, TRP-1, TRP-2, and microphthalmia-associated transcription factor (MITF) in a concentration-dependent manner. Nomilin significantly reduced the levels of p-CREB and p-PKA at the protein level and decreased the levels of skin-whitening-related factors MITF, tyrosinase, TRP-1, and TRP-2 at the mRNA level in a concentration-dependent manner. Thus, nomilin from yuzu seed husk can be used as a skin-whitening agent in cosmetics.

## 1. Introduction

Skin aging can be mainly divided into intrinsic and extrinsic aging. Intrinsic aging occurs when the bonds between the skin’s epidermis and dermis weaken, the ability of keratinocytes to divide declines, and the ability to form lipids decreases [1]. Extrinsic aging, also known as photoaging, is caused by long-term UV exposure and occurs when UV rays penetrate the epidermis, reaching deep into the dermis to damage collagen and elastin (elastic fibers), which maintain the dermis’ elasticity [2]. In response to these challenges, the skin overproduces reactive oxygen species (ROS), including superoxide anions and peroxides [3]. The increase in ROS activity damages DNA and increases transformation signals, ultimately increasing the level of transcription factor activator protein 1 [4]. UV radiation is directly involved in DNA mutagenesis, increases nuclear factor-κB levels, and decreases TGF-β levels [5]. These mechanisms affect the synthesis and degradation of collagen as well as the production of inflammatory cytokines. When the synthesis of collagen and elastin, both components of extracellular spaces, decreases due to UV exposure, the expression of various proteolytic enzymes of the extracellular matrix is promoted. The resulting lack of extracellular matrix proteins has been suggested to be the most important factor in photoaging [6].

Melanin, which determines the color of the skin, hair, and pupils, is produced in melanocytes (pigment cells) [7]. The process of melanin formation in melanocytes is called melanogenesis, and melanocytes are found in the lowermost layer of the skin epidermis. Melanin is a generic term for black or brown pigments in tissues, such as the skin and eyes, and mainly exists as melanin molecule that forms a strong bond with globulin. Melanin is an aggregate of small molecules and is categorized into eumelanin and pheomelanin based on race. Melanin moves through keratinocytes to the epidermis, thereby causing skin pigmentation [8]. Abnormal melanin synthesis mainly occurs through the oxidation of the precursor tyrosine due to excessive UV irradiation, disease, or genetic factors [9]. Tyrosine is oxidized to 3,4-dihydroxy-l-phenylalanine (DOPA) and DOPA quinone by tyrosinase in melanocytes, catalyzed to the intermediate DOPA chrome, and eventually polymerized to melanin via indole-5,6-quinone [10]. The inhibition of tyrosinase activity inhibits the biosynthesis of melanin polymer in the skin, and thus, is extremely important for the development of antioxidants and skin-whitening agents. Additionally, blocking melanosome transfer between melanocytes and keratinocytes can have a skin-whitening effect. Representative tyrosinase inhibitors derived from natural products include kojic acid, (a secondary metabolite from green mold), arbutin (isolated from bearberry leaves), and oxy-resveratrol (isolated from mulberry trees), in addition to other stilbene- and flavonoid-based compounds [11].

Kojic acid and arbutin—representative skin-whitening agents—have been used as additives in cosmetics. However, because they have various side-effects, including skin toxicity and allergies, there is an increasing demand for natural skin-whitening materials in the cosmetics industry. *Citrus junos* (yuzu) is mainly processed to manufacture yuzu syrup using the flesh and skin of yuzu, and due to its excellent flavor and unique acidic taste, is consumed as yuzu tea. However, 10–15% of yuzu fruit is left over as byproduct. Most byproducts remaining after the production of yuzu syrup and juice are discarded, with nearly 1800 tons discarded annually. Yuzu, which is mainly cultivated in northeast Asia, contains large amounts of vitamin C and flavonoids, such as hesperidin and limonin [12]. These flavonoids have been reported to have high anti-inflammatory and antioxidant activity. Flavonoids extracted from citrus species—in particular, hesperidin and naringin were reported to reduce the expression of inflammatory cytokines, such as tumor necrosis factor-α, interleukin-1, and interleukin-6 [13]. However, studies on the skin-whitening effect of yuzu seed byproducts are lacking. To our knowledge, no studies have been conducted on the mechanism of action of nomilin, a component found in yuzu seed byproducts, as a tyrosinase inhibitor and skin-whitening agent. Therefore, in this study, we isolated nomilin from discarded yuzu byproducts and evaluated its suitability as a tyrosinase inhibitor by performing in vitro kinetic and in vivo cell line experiments. In addition, its potential as a functional cosmetic ingredient with skin-whitening and anti-aging effects was discussed.

## 2. Materials and Methods

### 2.1. Chemicals

2,2′-Azino-bis (3-ethylbenzthiazoline-6-sulfonic acid) (ABTS), 2,2-diphenyl-1-picrylhydrazyl (DPPH), Folin–Ciocalteu reagent (for total phenolics), dimethyl sulfoxide-d_6_ (DMSO), mushroom tyrosinase (EC 1.14.18.1), and ascorbic acid were obtained from Sigma Aldrich (St. Louis, MO, USA). All reagents used were of analytical grade. Phospho-CREB (p-CREB) and phospho-PKA (p-PKA) were purchased from Cell Signaling (Danvers, MA, USA). Antibodies against tyrosinase (TYR), TRP-1, TRP-2, microphthalmia-associated transcription factor (MITF), CREB, PKA, α-melanocyte-stimulating hormone (α-MSH), and β-actin were purchased from Santa Cruz Biotechnology (Dallas, TX, USA). Horseradish peroxidase-conjugated anti-mouse, anti-goat, and anti-rabbit antibodies were purchased from Invitrogen (Carlsbad, CA, USA).

### 2.2. Chemical Extracts of Limonoid Aglycones and Limonoid Glucosides

We separated waste husk, shell, and meal from yuzu seeds. Two hundred grams of separated husk, shell, and meal were placed in 2 L of 100% ethanol at room temperature for 24 h for extraction [4,12]. The filtrate was concentrated using vacuum filtration. Then, 2 L of 100% ethanol was added to the residue, and extraction was repeated three times under the same conditions. Concentrated limonoid aglycones extracted from the husk, shell, and meal were named LA1, LA2, and LA3, respectively (Figure 1). After the limonoid aglycones were extracted, the residues were dried and placed in 2 L of water at 100 °C and 400 rpm for 2 h to extract limonoid glucosides. After extraction, the filtrate was concentrated by vacuum filtration; 2 L of water was added to the filtration residue, and extraction was repeated three times under the same conditions. Concentrated limonoid glucosides extracted from the husk, shell, and meal were named LG1, LG2, and LG3, respectively.

### 2.3. High-Performance Liquid Chromatography with Diode-Array Detection (HPLC–DAD) Analysis

The most effective extracts of limonoid aglycone (LA1, LA2, and LA3) and limonoid glucoside (LG1, LG2, and LG3) were analyzed quantitatively by HPLC. The HPLC system (DGU-20A; Shimadzu, Kyoto, Japan) consisted of an LC-20AD pump and a diode-array detector (SPD-20A). The compounds were identified by comparison with the retention times of standard materials [14].

### 2.4. Extraction of Nomilin

Medium-pressure liquid chromatography was performed using PREP UV-10V (Yamazen) and PUMP 582 (Yamazen, Osaka, Japan). The experiment was conducted under the following conditions: flow rate, 20 mL/min; UV detection, 254 nm; and column, ULTRA PACK (20 mm × 200 mm, 30 µm; Yamazen). The solvents used were *n*-hexane (A) and EtOAc (B). The gradient program for LA1 was (B) 100%, 0–10 min; (B) 90%, 10–50 min; (B) 80%, 50–90 min; (B) 70%, 90–105 min; (B) 50%, 105–120 min; (B) 0%, 120–125 min; and (B) 0%, 125–135 min. Nomilin was isolated by preparative HPLC with an X-Bridge Prep OBD C_18_ column (5.0 µm, 19 mm × 150 mm). Elution was performed with a linear gradient of methanol (0 min, 50/50; 30 min, 100/0; 100 min, 100/0) to obtain nomilin as a yellowish powder with 4.35% yield. ^1^H and ^13^C NMR spectra were recorded using DMSO-d_6_ and AVANCE III HD 400 MHz NMR (Bruker, Billerica, MA, USA) (Figure 2, Appendix A). Coupling constants were expressed in Hz, and chemical shifts were expressed on a d (ppm) scale. ^1^H NMR (DMSO-d_6_): δ = 1.03 (s, 3H, H-20), 1.15 (s, 3H, H-19), 1.31 (d, 1H, J = 9.32 Hz, H-8), 1.55 (s, 3H, H-18), 1.93 (s, 9H, H-21a, H-21b, H-23), 2.36 (m, 2H, H-12), 2.51 (m, 2H, H-11), 2.81 (dd, J = 16.20 Hz, J = 7.32 Hz, H-10), 3.12 (t, 2H, J = 14.4 Hz, H-5), 3.61 (d, 1H, J = 16.04 Hz, H-1), 3.93 (s, 1H, H-15), 4.84 (d, 1H, J = 7.12 Hz, H-3′), 5.46 (s, 1H, H-17), 6.51 (d, 1H, J = 1.20 Hz, H-3′), 7.67 (t, 1H, J = 1.68 Hz, H-4′), 7.73 (s, H-1′). ^13^C-NMR (DMSO-d_6_): δ = 15.72 (C-20), 16.77 (C-18), 17.04 (C-19), 20.88 (C-21a, C-21b), 23.05 (C-11), 31.95 (C-23), 33.72 (C-12), 35.77 (C-5), 37.47 (C-2), 44.06 (C-8), 44.25 (C-13), 50.90 (C-9), 52.51 (C-10), 53.26 (C-7), 66.19 (C-15), 71.40 (C-14), 77.88 (C-4), 79.63 (C-1), 85.00 (C-17), 110.65 (C-3′), 120.48 (C-2′), 142.13 (C-1′), 143.95 (C-4′), 167.58(C-22), 169.59 (C-3), 169.86 (C-16), 208.41 (C-6).

### 2.5. Antioxidant Activity Assay

#### 2.5.1. DPPH Radical Scavenging Activity

Radical scavenging activity was determined using the DPPH radical scavenging assay with some modifications [15]. We mixed 200 μL extract with 800 μL of 1 mmol/L methanolic DPPH. Mixtures were left for 15 min in the dark. Then, absorbance was measured at 517 nm with the SCINCO UV-Vis spectrophotometer (S-3100; Seoul, Korea). The scavenging activity of DPPH radicals was calculated using the following equation: Scavenging activity (%) = 100 × (A_0_ − A_1_)/A_0_, where A_0_ is the absorbance of the MeOH control and A_1_ is the absorbance in the presence of nomilin extracts. The inhibitory concentration (IC_50_) was defined as the amount of extract required for a 50% reduction of free radical scavenging activity. The IC_50_ values were obtained from the resulting inhibition curves. Results were compared with the activity of ascorbic acid (Sigma Aldrich, St. Louis, MO, USA) as a control.

#### 2.5.2. ABTS Radical Scavenging Activity

A 7 mM solution of ABTS was prepared in water. The ABTS stock solution was reacted with 7 mM potassium persulfate (final concentration), and the mixture was left at room temperature for 12–16 h before use to generate ABTS radicals. Radical scavenging was measured by mixing 200 µL of each sample and 1000 µL ABTS solution [16]. Mixtures were left for 15 min in the dark, and absorbance was measured at 730 nm with the SCINCO UV-Vis spectrophotometer S-3100 (Seoul, Korea). The scavenging activity of ABTS radicals was calculated using the following equation: Scavenging activity = 100 × (A_0_ − A_1_)/A_0_, where A_0_ is the absorbance of the water control and A_1_ is the absorbance in the presence of nomilin extract. IC_50_ values were obtained from the resulting inhibition curves. Results were compared with the activity of quercetin (Sigma Aldrich, St. Louis, MO, USA) as a control.

### 2.6. Total Polyphenol Content

Total polyphenol content in the fractionated samples was measured using a modified version of the Folin–Ciocalteu method [17]. A total of 500 μL of extract was mixed with 500 μL of Folin–Ciocalteu reagent and 500 μL of 2% sodium carbonate (*w/v*). The mixtures were left for 30 min at 25 °C. Absorbance was measured at 750 nm with a UV-Vis spectrophotometer (S-3100; SCINCO, Seoul, Korea). The extract samples were evaluated at a final concentration of 1 mg/mL. Total phenolic content was expressed as mg/mL of gallic acid equivalents (GAE) using the following equation, which was based on the calibration curve: *y* = 19.42*x* + 0.0541, *R*^2^ = 0.996, where *x* is the gallic acid equivalent (mg/g) and *y* is the absorbance.

### 2.7. Antimelanogenic Activity Assay

#### 2.7.1. Tyrosinase Inhibition Assay

The tyrosinase inhibition assay was performed according to Macrini et al. [18], with a few modifications. We used 1250 U/mL of tyrosinase (Sigma Aldrich, St. Louis, MO, USA) for the experiment. We added 10 μL of tyrosinase to the wells of 96-well microplates. Then, 70 μL of pH 6.8 phosphate buffer solution and 60 μL of nomilin (10–200 μg/mL), LA1 (10–500 μg/mL), LG1 (10–500 μg/mL), and ascorbic acid (10–100 μg/mL) as a standard were added to the mixture in order. Next, 70 μL of L-tyrosine (Sigma Aldrich) was added at a concentration of 0.3 mg/mL in distilled water (the final volume in the wells was 210 μL). The absorbance of the microplate wells was read using a spectrophotometer (Synergy HT; BIO-TEX, Winooski, VT, USA) at 510 nm (T_0_). The microplates were incubated at 30 °C ± 1 °C for 60 min, and absorbance was measured (T_1_). The microplates were further incubated for 60 min at 30 °C ± 1 °C, and absorbance was measured (T2). The inhibitory percentages at the two timepoints (T_1_ and T_2_) were obtained according to the following formula: Inhibition activity (IA)% = (C − S)/C × 100, where IA% is the inhibitory activity, C is the absorbance of the negative control, and S is the absorbance of the sample or positive control (absorbance at time T_1_ or T_2_ minus absorbance at time T_0_) [19].

#### 2.7.2. Enzyme Kinetic Assay

Tyrosinase (EC 1.10.3.1) is an enzyme that converts L-tyrosine to DOPA and finally to DOPA quinone. To evaluate inhibition, L-DOPA was used as a substrate at concentrations of 0.5, 1.0, 1.5, and 2.0 mM. Tyrosinase inhibition was detected using a spectrophotometer (Synergy HT; BIO-TEX, Winooski, VT, USA). The IC_50_ assay was performed for tyrosinase according to Fan et al. [20].

For the test, 20 µL aliquots of a solution composed of 500 U/mL mushroom tyrosinase (Sigma Aldrich, St. Louis, MO, USA) were added to 96-well microplates. Then, 100 µL of pH 6.8 phosphate buffer solution and 60 µL of nomilin (0.2–1.0 mM) were added. Absorbance was measured at 510 nm (T0) using a microplate reader (Synergy HT; BIO-TEX, Winooski, VT, USA). The microplates were incubated at 30 °C ± 1 °C for 30 min, and the absorbance was measured again (T1). The microplates were further incubated for 30 min at 30 °C ± 1 °C, and absorbance was measured (T2). The inhibitory percentages at the two timepoints (T1 and T2) were obtained based on the following formula: IA% = (C − S)/C × 100, where IA% is the inhibitory activity, C is the negative control absorbance, and S is the absorbance of the sample or positive control (the absorbance at time T0 subtracted from the absorbance at time T1 or T2) [21].

### 2.8. Molecular Docking Procedure

Molecular docking was performed to predict the binding site of mushroom tyrosinase and TRP-1 to nomilin using the Glide module in the Schrodinger package [22,23]. The X-ray crystal structures of tyrosinase (PDB ID: 2Y9X) and TRP-1 (PDB ID: 5M8O) were retrieved from the Protein Data Bank (http://www.rcsb.org (accessed on 10 October 2020)). The retrieved protein structures were processed using Protein Preparation Wizard in the Schrodinger package to remove crystallographic water molecules, add hydrogen atoms, and assign protonated states and partial charges. The missing side chains and loops were built and refined using the Prime tool of the Schrodinger suite [24]. All protein residues were parameterized using the OPLS3e force field [25,26]. Finally, restrained minimization was performed until the converged average root mean square deviation of heavy atoms was 0.3 Å. Binding mode predictions of nomilin with mushroom tyrosinase and TRP1 were performed using the Glide docking tool in the Schrodinger package. Docking grid boxes were generated considering the catalytic sites of mushroom tyrosinase and TRP-1. Nomilin was docked into the catalytic site of each protein using standard precision scoring modes. The 3D structure of nomilin was minimized using the Macromodel module of the Schrodinger suite.

### 2.9. Cell Culture

B16F10 melanoma cells, a murine melanoma cell line, were purchased from the Korea Cell Line Bank (KCLB80008, Seoul, Korea). Cells were maintained in Dulbecco’s modified Eagle’s medium (DMEM; HyClone, MA, USA) supplemented with 10% fetal bovine serum (HyClone), 50 units/mL penicillin, and 50 µg/mL streptomycin at 37 °C in a humidified atmosphere with 5% CO_2_ at 37 °C.

### 2.10. MTT Cell Viability Assay

Cell viability analysis was performed using the 3-(4,5-dimethylthiazol-2-yl)-2,5-diphenyltetrazolium bromide (MTT) assay. B16F10 cells were cultured at 1 × 10^4^ cells/cm^3^ in six-well plates. After 24 h, the cells were treated with 25, 50, or 100 µg/mL nomilin for 48 h. At the end of incubation, 100 µL of MTT solution (1 mg/mL in DMEM) was added to each well. After incubation at 37 °C for 1 h, the medium was gently removed, and 400 µL of DMSO was added. The absorbance of each well was measured at 570 nm using a spectrophotometer.

### 2.11. Measurement of Melanin Content

Melanin content was determined according to Hosoi et al. [27]. B16F10 cells were cultured at 1 × 10^4^ cells/cm in six-well plates. After 24 h, the cells were stimulated with 1 µg/mL α-MSH. Simultaneously, various concentrations of nomilin (62 and 125 µM) were added for 48 h. Then, the cells were washed with phosphate-buffered saline (PBS) and harvested after trypsin treatment. The collected cells were suspended in 100 µL of 1 N NaOH, and absorbance was measured at 405 nm using a spectrophotometer.

### 2.12. Measurement of Intracellular ROS Generation

ROS generated by *t*-BHP as evaluated using 2,7-dichlorodihydrofluorescein (H2DCF-DA) [28]. H2DCF-DA is oxidized to a green, highly fluorescent compound called 2,7-dichlorofluorescein (DCF) upon ROS generation. B16F10 cells were treated with various concentrations of nomilin (10–100 μg/mL) for 24 h. Then, the cells were rinsed with PBS and incubated with 100 μM H2DCF-DA for an additional 30 min at 37 °C. A fluorescence plate reader (Synergy HT; BIO-TEX, Winooski, VT, USA) was used to measure DCF fluorescence intensity at Ex./Em. = 488/525 nm. The experiments were performed three times. The values were expressed as percentage fluorescence relative to control.

### 2.13. Immunoblotting

B16F10 cells were treated with different concentrations of nomilin, disrupted using lysis buffer containing protein inhibitors, and frozen for 24 h in a deep-freezer. The frozen cells were thawed on ice for ~90 min and vortexed 3 to 6 times to disrupt the cells for protein extraction. SDS immunoblotting and polyacrylamide gel electrophoresis were performed as described [29], with a few modifications. Briefly, the samples were separated on 7.5% SDS-PAGE and transferred to a nitrocellulose membrane electrophoretically. The membrane was first incubated with the primary antibodies, and then with horseradish peroxidase-conjugated secondary antibodies. The signal was detected using the Enhanced Chemiluminescence Detection Reagent (Amersham Biosciences, Little Chalfont, UK). β-actin was used as the loading control.

### 2.14. RNA Isolation and Reverse-Transcription Polymerase Chain Reaction (RT-PCR)

Total RNA was extracted using TRIzol reagent (Invitrogen), in accordance with the manufacturer’s instructions. To obtain cDNA, total RNA (2 µg) was reverse-transcribed using an oligo(dT) primer. The cDNA was amplified using the High-Capacity cDNA synthesis kit (Bioneer, Daejeon, Korea) in a PCR machine (Bio-Rad, Hercules, CA, USA). PCR was performed using a PCR premix (Bioneer), and real-time RT-PCR was performed using the StepOne model (Applied Biosystems, Foster City, CA, USA) and SYBR Green premix, according to the manufacturer’s instructions (Applied Biosystems). Primers were synthesized by Bioneer. The following primer sequences were used: mouse tyrosinase: 5′-ATAACAGCTCCCACCAGTGC-3′ (sense) and 5′-CCCAGAAGCCAATGCACCTA-3′ (antisense); mouse MITF: 5′-CTGTACTCTGAGCAGCAGGTG-3′ (sense) and 5′-CCCGTCTCTGGAAACTTGATCG-3′ (antisense); and mouse TRP-1: 5′-AGACGCTGCACTGCTGGTCAAGCCTGTAGCCCACGTCGTA-3′ (sense) and 5′-GCTGCAGGAGCCTTCTTTCT-3′ (antisense). The expression of glyceraldehyde 3-phosphate dehydrogenase was used as an endogenous control for qRT-PCR [30].

### 2.15. Statistical Analysis

For statistical analysis, we used IBM SPSS online version 26.0 (SPSS, Inc., Chicago, IL, USA). For differences between groups, one-way analysis of variance was used, and statistical significance was evaluated. In addition, for each statistically significant effect of treatment, Duncan’s multiple range test was used for comparing multiple means. Data were expressed as mean ± standard deviation (SD).

## 3. Results and Discussion

### 3.1. Yuzu Seed Isolation and Yield

Limonoid compounds are secondary metabolites and triterpene derivatives that are mainly present in mature fruits and seeds. A total of 38 limonoid aglycones: 23 neutral limonoids and 15 acidic limonoids, have been isolated from various fruits. Recently, 36 aglycones and 20 glucosides were isolated from limonoids [31]. In citrus fruits, limonoid compounds exist in the form of aglycones or glucosides, with limonin and nomilin being the most abundant [32]. Notably, citrus seeds contain abundant limonoid compounds, with more aglycones than glucosides [4]. In this study, we separated yuzu seed byproducts into husk, shell, and meal and compared results of their composition and material balance. Comparison of material balance (in 200 g each of husk, shell, and meal) revealed that the yields of yuzu seed limonoid aglycones were 12.35, 11.16, and 7.36 g, respectively; yuzu seed husk (LA1) had the highest yield (Figure 1). In another study, yuzu seeds were classified into three types, and their harvest rates were measured. Water content was lowest in yuzu seed. Moreover, high-cost limonoids and yuzu seed oil with high antioxidant activity were extracted from waste yuzu seeds, which had fat-soluble limonoid aglycone (330.6 mg g^−1^ of dry seeds), water-soluble limonoid glucosides (452.0 mg g^−1^ of dry seeds), and oil (40 mg g^−1^ of green seeds) [12]. In another study, yuzu seeds were separated into three parts to measure harvest rate. The findings were consistent with ours, with the shells having the highest yield [33]. Thus, our findings are consistent with those of others—yuzu seed aglycones had the highest yield; the high content of limonoid aglycones was due to the relatively lower water content than that in other parts.

### 3.2. Antioxidant Activity of Yuzu Seed Extracts

#### 3.2.1. DPPH Radical Scavenging Activity

Radicals with an odd number of electrons are highly energetic and unstable. They are highly reactive and cause oxidative reactions in vivo, resulting in cell and tissue damage. Lipid peroxidation due to a radical chain reaction is known to be the main cause of accelerated skin aging. Free radicals take electrons from reducing substances and become reduced. The strength of the reducing power is important in protecting skin cells against oxidative damage. The reducing power of antioxidants can be evaluated using the radical scavenging activity assay [34]. Evaluation of the DPPH free radical scavenging ability of limonoid aglycone and limonoid glucoside revealed that the antioxidant activities of limonoid aglycone and limonoid glucoside are similar (Appendix A). IC_50_ values of limonoid aglycones (LA1, LA2, and LA3) and limonoid glucosides (LG1, LG2, and LG3) were 942.02, 1250.08, and 1240.15 and 1121.84, 1302.20, and 1102.31 μg/mL, respectively (Table 1). The activity of LA1 was highest. In previous studies, the radical scavenging activities of 70% ethanol extracts of yuzu shells and the ethyl acetate fraction were 512.1 and 514.0 μg/mL, respectively [34]. In another study, the sample volume of the free magnetic seed chamber ethanol extract ranged from 20.65% to 57.94% when the sample volume ranged from 100 to 1000 μg/mL, respectively [35]. The electron-donating ability of the citron juice showed 80% or more when a sample solution having a concentration of 0.1% was added at a concentration of 1 × 10^4^ M DPPH, and an interaction of phenol, hesperidin, and naringin, among others, as active ingredients was reported [36].

#### 3.2.2. ABTS Radical Scavenging Activity

The radical scavenging ability is an index of the antioxidant activity of phenolic substances, such as phenolic acid and flavonoids. Greater reducing power indicates higher electron-donating ability [36]. The activity of LA1 and LG1 extracted from the seed coat was the highest (Table 1 and Appendix A). In another study, the ABTS radical scavenging activity of 70% ethanol extracts of yuzu seeds was found to be below 150 μg/mL [37]. The ABTS radical scavenging activities of limonin and nomilin extracted from sun-dried pomelo seeds were found to be 201.33 and 346.47 μg/mL, respectively [38]. Furthermore, the antioxidant activity of an extract obtained from yuzu seed husk was found to be high. This finding is consistent with the results of our study. Polyphenols, which have high antioxidant activity, were extracted in addition to limonoid aglycones and limonoid glucosides. In addition, the antioxidant activity of yuzu seeds was determined to be higher than that of yuzu peel, probably due to the relatively high content of flavonoids, such as hesperidin, in yuzu seeds.

#### 3.2.3. Total Polyphenol Content

Polyphenolic substances confer special color to plants and act as substrates in redox reactions. Polyphenolic substances, including flavonoids and tannins, are aromatic compounds, with two or more phenol hydroxyl groups in one molecule [39]. Polyphenols play various physiological roles, such as preventing tooth decay; suppressing hypertension; and exerting anti-AIDS, antioxidant, and anti-cancer effects. Our findings revealed that total polyphenol content was consistent with antioxidant activity. The content of LA1 was the highest (Table 1). Furthermore, polyphenol content in limonoid aglycones and limonoid glucosides extracted from yuzu seed husk was high. In a study, the total polyphenol content of *n*-hexane and 70% ethanol extracts of citron seeds were reportedly 201.84 and 246.31 GAE mg/100 g, respectively [38,40]. Consistently, Woo et al. reported [39] a total polyphenol content of 5.67 GAE mg/g. The extraction yield of citron seeds with 75% ethanol was 9.82%, and the total phenol content of the crude extract was 24.44 GAE mg/100 g.

Nomilin has a seven-membered oxepin ring, and its main functional groups are dilactone and acetic ester. The yuzu seed husk used in this study contains an abundance of components, such as naringin, hesperidin, and chlorogenic acid, and thus, is believed to have high antioxidant activity [41]. Antioxidant activity varies depending on the presence or absence of hydroxyl groups. Although nomilin has a small number of hydroxyl groups, the presence of functional groups, such as dilactone and acetic ester ensures that the electron-attractive force is strong and radical scavenging ability is reduced through interactions, such as covalent bonds, hydrogen bonds, and van der Waals’ forces.

### 3.3. HPLC Analysis of Yuzu Seed Parts

According to previous research, citrus fruits, such as yuzu, contain an abundance of flavonoid compounds, such as hesperidin and naringin, in the peel [42]. Levels of vitamin C, vitamin D, and minerals in yuzu are more than three times higher than those in lemon [43]. In this study, we first extracted and separated the hydrophilic (limonoid glucoside) and hydrophobic (limonoid aglycone) components from yuzu seed byproducts. Then, the limonin and nomilin contents in limonoid aglycone of each part of yuzu seeds (husks, shells, and meal) were detected and quantitatively analyzed by HPLC. The results showed that the limonin contents in LA1 (yuzu seed husks), LA2 (yuzu seed shells), and LA3 (yuzu seed meal) were 641.4, 315.5, and 595.1 mg/g, respectively, and the corresponding nomilin contents were 538.7, 690.7, and 1725.8 mg/g, respectively (Appendix A, Appendix A). HPLC analysis of polyphenol compounds in LG1, LG2, and LG3 revealed the presence of eight standard substances (Appendix A, Appendix A). The levels of chlorogenic acid, naringin, and hesperidin in LG1 were 409.35, 430.17, and 436.76 mg/g, respectively. The levels of these compounds were also high in LG2 and LG3, but lower than that in LG1. These findings explain the difference in antioxidant activity between these fractions. Notably, the higher content of polyphenols in the yuzu seed husk extract explains its high antioxidant activity. In a study, HPLC analysis of yuzu seed extract revealed that levels of naringin, hesperidin, limonin, and nomilin were 100.43, 21.78, 170.98, and 45.36 mg/100 g, respectively [14]. In this study, we detected 3–4 times more polyphenols than other studies, probably because we divided yuzu seeds into three parts.

### 3.4. Nomilin Tyrosinase Inhibitory Activity

For testing skin-whitening efficacy in vitro, the in vitro tyrosinase inhibition assay and the in vitro DOPA oxidation inhibition assay are widely used. Tyrosinase is a copper-containing polyphenol oxidase that catalyzes the hydroxylation of monophenols. It is found in microorganisms and animal and plant tissues and contributes to the synthesis of melanin and the production of pigments [44]. Tyrosinase is involved in the initial rate-determining step of the melanin biosynthesis pathway in humans. Many skin-whitening products function by inhibiting tyrosinase. The in vitro tyrosinase inhibition assay evaluates the degree of tyrosinase inhibition in vitro [45]. The DOPA oxidation assay evaluates the effect of skin-whitening compounds by measuring the inhibition of tyrosinase activity, which catalyzes the rate-determining step of the melanin biosynthesis. We isolated and evaluated the potential of nomilin as a skin-whitening agent by reviewing the tyrosinase inhibitory activity of nomilin, which is abundant in the aglycone fraction of yuzu seeds. To our knowledge, this has not previously been attempted. First, based on the antioxidant activity results, the tyrosinase inhibitory activities of LA1 and LG1, which were the highest, were compared. The tyrosinase inhibitory activity of nomilin, the most abundant component in LA1, was confirmed (Figure 3). For the positive control, we used ascorbic acid, a known skin-whitening agent. The IC_50_ values of LA1, LG1, nomilin, and ascorbic acid were 192.63, 688.53, 87.17, and 38.71 μg/mL, respectively. Nomilin showed tyrosinase inhibitory activity higher than that of the extract and lower than that of ascorbic acid. In a study, the tyrosinase inhibitory activities of blue yuzu peel and yellow yuzu peel were compared, and tyrosinase inhibitory activity was found to be related to the content of phenol compounds in yuzu peel [46]. Two types of limonoid aglycones are present in the seed shell, namely, monolactones, such as limonic acid A-ring lactone, and dilactones, such as limonin [47]. Nomilin, isolated from yuzu seed shells, is a limonoid compound with a phenolic structure, and due to its oxidation inhibitory function, is estimated to have high tyrosinase inhibitory activity. In addition, studies have reported that compounds with a furan structure in coffee byproducts exhibited high antioxidant activity [48]. Nomilin extracted from yuzu seed byproducts has a furan structure and an acetate group; therefore, it is proposed to exhibit high antioxidant activity and tyrosinase inhibitory activity.

### 3.5. Enzyme Kinetic Analysis

The number of active sites is constant; thus, at high substrate concentrations, enzyme saturation occurs. At high substrate concentrations, the rate of enzyme reactivity is thus independent of substrate concentration; however, at low substrate concentrations, it is proportional to substrate concentration. In a multistep enzymatic reaction, the reaction with the largest K_m_ value determines the reaction rate of the entire reaction system. In this study, we calculated K_m_ and V_max_ and identified the type of inhibition using the Lineweaver–Burk equation (Figure 4). The pattern of inhibition of an enzyme depends on the binding site and the type of binding mode. During competitive inhibition, the inhibitor competitively binds to the substrate noncovalently, thereby inhibiting enzyme activity. In noncompetitive inhibition, the inhibitor reversibly binds to both the free enzyme and the enzyme–substrate complex to exhibit inhibitory effects. In this study, values of K_m_, V_max_, and the inhibition constant (K_i_) of nomilin against tyrosinase were 0.5049 mM, 0.3931 mmol/min, and 0.6408 mM, respectively. The Lineweaver–Burk plot was linear, confirming that the kinetic behavior was noncompetitive.

### 3.6. Molecular Docking Study

Molecular docking is used to predict binding by modeling the binding and interaction of proteins and ligands at the atomic level [23]. We used molecular docking to predict the binding between nomilin and mushroom tyrosinase and TRP-1 (Figure 5).

Molecular docking predicted that nomilin interacts with Tyr 65, Tyr 78, Ala 323, Glu 322, and Asn 81 adjacent to the catalytic site of mushroom tyrosinase. Similarly, nomilin was predicted to interact with Lys 334, Arg 196, Gln 378, and Gln 376 around the catalytic site of TRP-1. Thud, nomilin is predicted to bind adjacent to the catalytic sites of mushroom tyrosinase and TRP-1 (Figure 5c,d).

### 3.7. Cytotoxicity Evaluation and Quantitative Analysis of ROS

Cell viability and cytotoxicity were measured colorimetrically using the MTT reagent, which turns purple when mitochondrial dehydrogenase and MTT tetrazolium react during cellular metabolism to form MTT formazan rehydrated with DMSO [49]. For each concentration (25–100 µg/mL), the cytotoxicity of nomilin was determined to be <100 µg/mL (Figure 6a). The concentration that did not show the toxicity of nomilin was confirmed to be ≤25 µg/mL, which is lower than the result of this study [50]. Another study evaluated the toxicity of citron peel extract and reported that it was safe at <800 µg/mL. In this study, nomilin was isolated from the yuzu seed husk, and the non-toxic concentration was found to be <100 µg/mL [51]. These results suggest that the range of toxicity may appear differently depending on the yuzu seed extraction method and cell types. Flow cytometry was performed using the fluorescent probe DCF-DA to confirm the effect of nomilin on the reduction of the total amount of intracellular ROS generated during metabolism. B16F10 cells were pretreated with nomilin for 24 h and then with 500 mM *t*-BHP, and the following results were obtained (Figure 6b). When nomilin was applied at concentrations of 10, 25, 50, and 100 μg/mL, the total amount of ROS was reduced by 17%, 23%, 45%, and 84%, respectively.

### 3.8. Melanin Content and Cell Morphology in Nomilin-Treated B16F10 Cells

B16F10 melanoma cells were treated with α-MSH (1 μg/mL). Nomilin and limonin isolated from yuzu seeds were added to the medium. Cells were cultured for 48 h, and secreted melanin was measured. The findings revealed that melanin content decreased in a concentration-dependent manner. The skin-whitening activity of nomilin was similar to that of arbutin, a positive control, at 100 μg/mL (Figure 7). Morphologically, B16F10 melanocytes showed a pattern consistent with the results of melanin content. In a study measuring the melanin content of yuzu seed extracts in B16F10 melanoma cells, strong skin-whitening activity was recorded at 0.02% [21]. In another study, a significant decrease in melanin content was seen (1.85-fold) in melanocytes treated with nomilin compared with those treated with arbutin (positive control) [52]. Additionally, melanin content was decreased by yuzu peel extract, which showed better activity than kojic acid, as a positive control, at a concentration of 0.02%. Most studies have not measured melanin content or tyrosinase inhibitory activity. In this study, we went one step further and investigated the skin-whitening activity in B16F10 melanoma cells.

### 3.9. Effect of Nomilin on Anti-Melanogenesis-Related Proteins in B16F10 Cells

UV rays, cytokines, growth factors, and hormones regulate melanogenesis. In this context, α-MSH is an important hormone [53]. α-MSH is secreted from the middle of the pituitary gland and binds to melanocortin 1 receptor, a membrane receptor expressed only in melanocytes, which activates adenylyl cyclase. This in turn amplifies the intracellular cAMP signal, induces protein kinase A (PKA) activation, and increases the expression of MITF (a transcription factor specific to melanocytes) by activating intracellular cAMP response element binding protein (CREB) [54]. Melanin is synthesized through intracellular signaling mechanisms, among which cAMP/PKA is the main pathway. MITF promotes the transcription of tyrosinase, TRP-1, and TRP-2 during melanin synthesis through CREB [55]. Against this background, the key step in skin-whitening is to inhibit tyrosinase, an enzyme involved in melanin biosynthesis, and to inhibit the synthesis of proteins involved in upstream and downstream signaling processes of the melanin synthesis pathway [56].

α-MSH treatment increased the levels of tyrosinase, TRP-1, and TRP-2, whereas nomilin treatment significantly decreased the levels of tyrosinase, TRP-1, and TRP-2 in B16F10 cells (Figure 8). Moreover, induction of MITF expression by 1 µg/mL α-MSH was inhibited by treatment with 25–100 µg/mL nomilin (Figure 8c,d). Thus, nomilin inhibited melanogenesis by downregulating MITF signaling. As shown in Figure 8e, nomilin preincubation inhibited α-MSH-induced phosphorylation of protein kinase and cAMP response element binding protein (PKA/CREB). Thus, the suppressive mechanism of nomilin was related to the inhibition of PKA/CREB signaling.

### 3.10. Effect of Nomilin on Anti-Melanogenesis-Related Genes in B16F10 Cells

To verify the skin-whitening activity of nomilin, we evaluated the expression of related genes, namely, TYR, TRP-1, TRP-2, and MITF, at the mRNA level using RT-PCR (Figure 9). The results revealed that the expression of genes (tyrosinase, TRP-1, and MITF) related to melanin production via nomilin was induced by α-MSH, whereas nomilin treatment resulted in significantly decreased mRNA expression (25–100 µg/mL).

In particular, nomilin was more effective in inhibiting mRNA expression at 100 µg/mL than at other concentrations. Moreover, nomilin showed more potent inhibitory activity than the positive control arbutin.

## 4. Conclusions

In this study, we separated and extracted limonoid and aglycone fractions from yuzu seed byproducts and confirmed that nomilin is a novel tyrosinase inhibitor. Nomilin was separated from the aglycone fraction as a single substance with strong antioxidant and skin-whitening activities. These effects were mediated by the activation of the PKA/CREB signaling pathway involved in melanogenesis in B16F10 cells. The results of this study suggest that the use of natural ingredients in the development of new skin-whitening materials can resolve the problems associated with using chemicals. Nomilin from yuzu seeds directly inhibited tyrosinase and was involved in protein synthesis and transcription factor regulation during the skin-whitening signal transduction mechanism. Therefore, nomilin has a high potential as a novel natural skin-whitening agent; however, more comprehensive molecular studies are needed to confirm its ability to reduce melanin pigmentation.

## Figures and Tables

**Figure 1 antioxidants-11-01636-f001:**
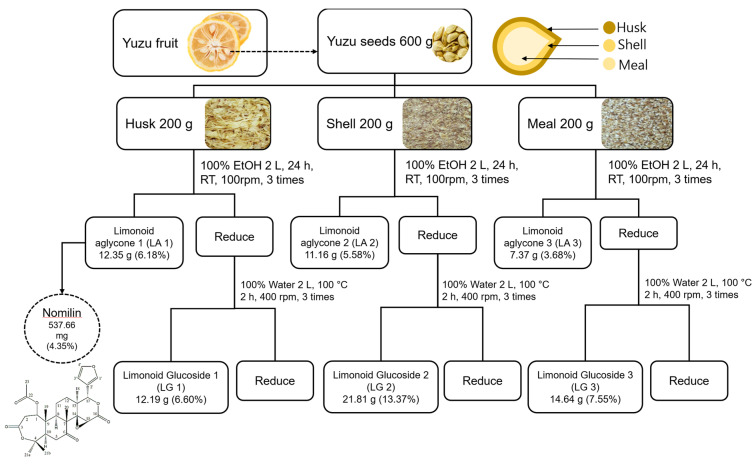
Schematic diagram showing the preparation of yuzu seed extracts. Percentages in parenthesis mean extraction yields based on the initial quantity of yuzu seeds. LA: limonoid aglycone, LG: limonoid glucoside.

**Figure 2 antioxidants-11-01636-f002:**
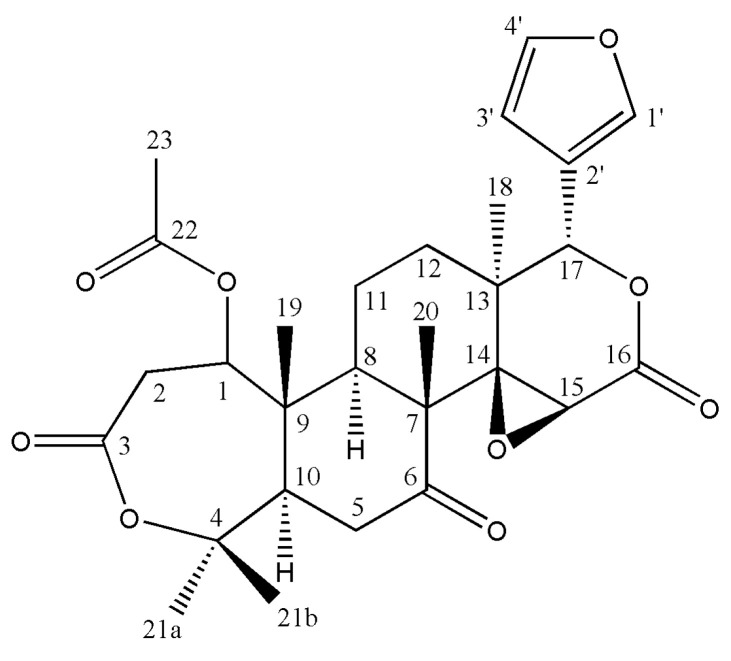
Structure of nomilin {7-(furan-3-yl)-1,8,12,17,17-pentamethyl-5,15,20-trioxo-3,6,16-trioxapentacyclo [9.9.0.0^2^,^4^.0^2^,^8^.0^12^,^18^]icosan-13-yl acetate} derived from yuzu seed husk.

**Figure 3 antioxidants-11-01636-f003:**
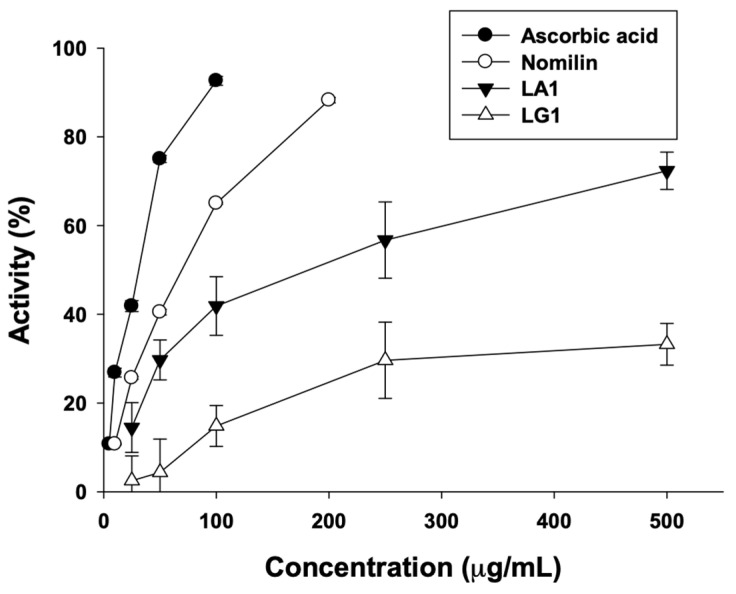
The inhibitory effect of ascorbic acid, nomilin, LA1, and LG1 on mushroom tyrosinase in a cell-free system. LA: limonoid aglycone, LG: limonoid glucoside.

**Figure 4 antioxidants-11-01636-f004:**
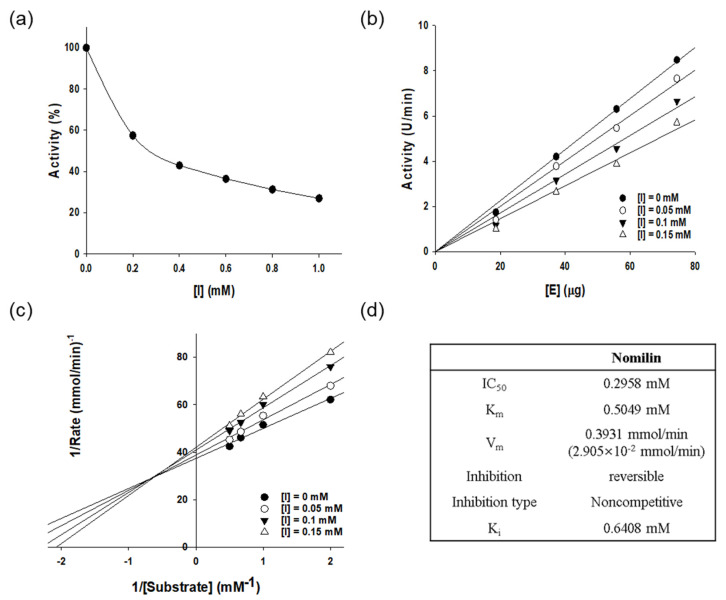
(**a**) Effect of nomilin on the activity of mushroom tyrosinase during the catalysis of L-DOPA (enzyme concentration 4.0 µg/mL). (**b**) Relationship between catalytic activity of mushroom tyrosinase and enzyme concentration at different concentrations of nomilin. Concentrations of nomilin for curves 1-4 were 0, 0.05, 0.1, and 0.15 mM, respectively. (**c**) Lineweaver-Burk plots for the nomilin-mediated inhibition of mushroom tyrosinase during the catalysis of DOPA at 30 °C and pH 6.8. Concentrations of quercetin for curves 1-4 were 0, 0.05, 0.1, and 0.15 mM, respectively; enzyme concentration was 4.0 µg/mL. The inset represents the plot of Kmapp versus quercetin concentration for determining the inhibition constant KI. (**d**) Kinetic parameters and microscopic inhibition rate constants of mushroom tyrosinase in the presence of nomilin.

**Figure 5 antioxidants-11-01636-f005:**
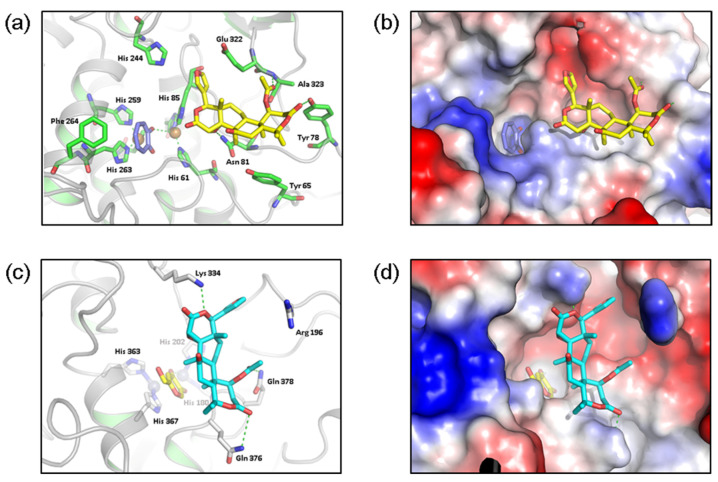
Molecular docking study. (**a**) Binding model of mushroom tyrosinase and nomilin; (**b**) surface model of mushroom tyrosinase and nomilin; (**c**) binding model of human tyrosinase and nomilin; (**d**) surface model of human tyrosinase and nomilin.

**Figure 6 antioxidants-11-01636-f006:**
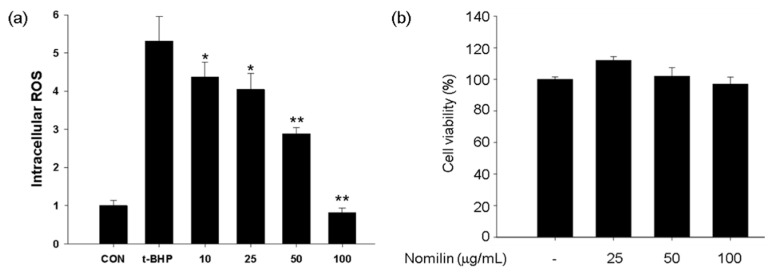
Measurement of (**a**) intracellular ROS production in and (**b**) viability of B16F10 melanoma cells treated with nomilin. Nomilin: 25–100 μg/mL (48.59–194.34 μM). * *p* < 0.05, ** *p* < 0.01, compared with t-BHP treatment.

**Figure 7 antioxidants-11-01636-f007:**
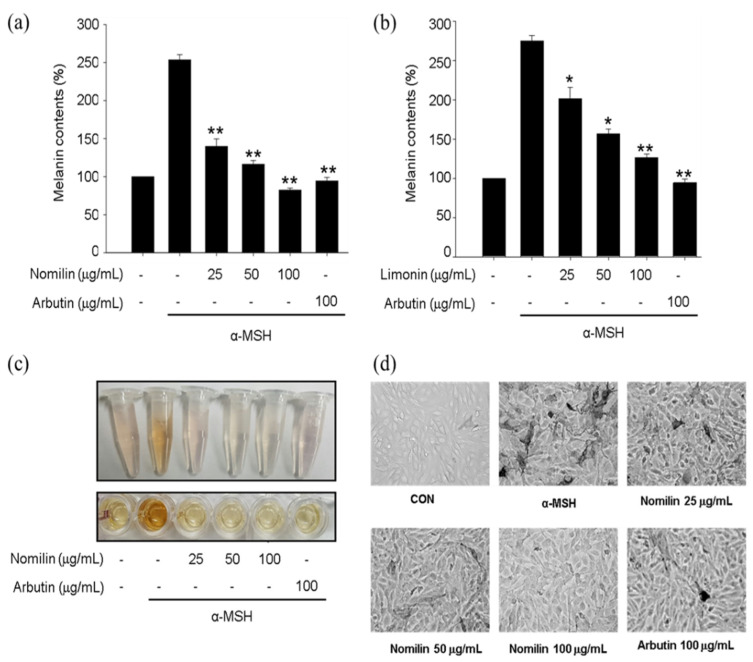
Measurement of melanin content in B16F10 melanoma cells treated with nomilin and limonin. (**a**) Measurement of melanin content with 25–100 μg/mL nomilin (48.59–194.34 μM). (**b**) Measurement of melanin content with 25–100 μg/mL limonin (53.13–212.53 μM). Relative melanin content was determined at 72 h after treatment. *n* = 3, error bars, mean ± standard deviation. Effect of 100 μg/mL arbutin (367 μM) on melanin synthesis and tyrosinase activity in B16F10 cells. (**c**) B16F10 melanoma cells. (**d**) Morphological changes due to nomilin. B16F10 cells were cultured for 48 h in the presence of 25–100 μg/mL (48.59–194.34 μM) nomilin and 100 μg/mL (367 μM) arbutin as a positive control or 1 μg/mL α-MSH. Significantly different compared with α-MSH, * *p* < 0.05, ** *p* < 0.01.α-MSH: α-melanocyte-stimulating hormone.

**Figure 8 antioxidants-11-01636-f008:**
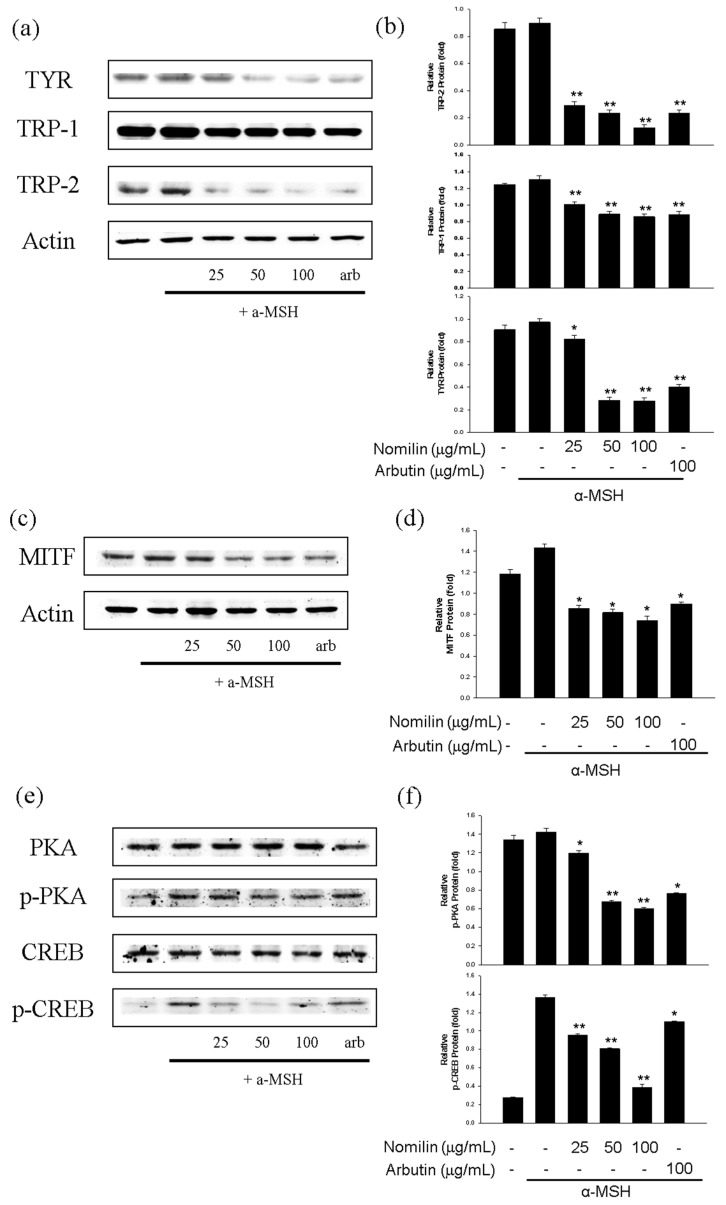
(**a**) Effects of nomilin and arbutin on tyrosinase, TRP-1, and TRP-2 expression in B16F10 melanoma cells. B16F10 cells were treated with different concentrations of nomilin and arbutin prior to α-MSH treatment for 24 h. β-actin was used as a loading control antibody. (**b**) Quantitative analysis of tyrosinase, TRP-1, and TRP-2 by Western blotting. Cell lysates were subjected to Western blotting using antibodies against tyrosinase, TRP-1, and TRP-2. (**c**) Effects of nomilin and arbutin on MITF expression in B16F10 cells. B16F10 cells were treated with the indicated concentrations of nomilin and arbutin prior to α-MSH treatment for 4 h. (**d**) Quantitative analysis of MITF by Western blotting. (**e**) Effects of nomilin and arbutin on p-CREB, CREB, p-PKA, and PKA levels in B16F10 cells. B16F10 cells were treated with the indicated concentrations of nomilin and arbutin prior to α-MSH treatment for 3 h. (**f**) Quantitative analysis of p-CREB, CREB, p-PKA, and PKA by Western blotting. Values are significantly different with Duncan’s multiple range test (significant compared with the vehicle-treated control, * *p* < 0.05, ** *p* < 0.01, bars indicate SD). CREB: cAMP response element binding protein, p-CREB: phospho-CREB, α-MSH: α-melanocyte-stimulating hormone, MITF: microphthalmia-associated transcription factor, and PKA: protein kinase A, p-PKA: phospho-PKA.

**Figure 9 antioxidants-11-01636-f009:**
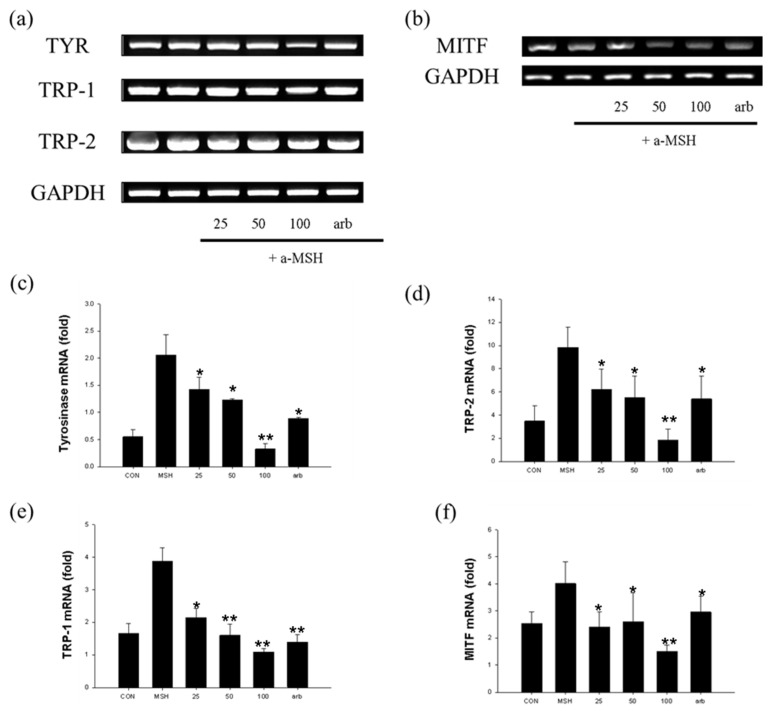
RT-PCR analysis of nomilin. (**a**) TYR, TRP-1, and TRP-2 transcripts were analyzed using RT-PCR. The level of GAPDH mRNA was used as control. (**b**) MITF transcripts were analyzed using RT-PCR. The level of GAPDH mRNA was used as control. (**c**) Effects of nomilin on tyrosinase, (**d**) TRP-1, (**e**) TRP-2, and (**f**) MITF mRNA in B16F10 cells. Data from separate experiments are presented (statistically significant compared with the vehicle-treated control, * *p* < 0.05, ** *p* < 0.01, bars indicate SD). α-MSH: α-melanocyte-stimulating hormone, MITF: microphthalmia-associated transcription factor, TYR: tyrosinase.

**Table 1 antioxidants-11-01636-t001:** IC_50_ values of the radical scavenging activity and total polyphenol content of limonoid aglycone and limonoid glucoside.

Sample	DPPH IC_50_ (μg/mL)	ABTS IC_50_ (μg/mL)	TPC (GAE mg/g)
LA1	942.02 ± 45.91 ^a,b^	626.34 ± 14.68 ^a,b^	6.72 ± 0.14
LA2	1250.08 ± 96.53	808.86 ± 13.25 ^a,b^	5.38 ± 0.32
LA3	1240.15 ± 116.90	668.41 ± 52.53	2.05 ± 0.04
LG1	1121.84 ± 31.57 ^a,b^	844.78 ± 13.96 ^a,b^	6.17 ± 0.23
LG2	1302.20 ± 22.91 ^a,b^	972.49 ± 14.38 ^a,b^	4.99 ± 0.63
LG3	1102.31 ± 18.80 ^a,b^	1077.79 ± 18.18 ^a,b^	1.87 ± 0.06
Nomilin	53.37 ± 0.83	57.9 ± 0.75	-
Ascorbic acid	56.97 ± 1.27	39.95 ± 0.73	-

Each value is the mean ± standard deviation (SD) of three experiments. Differences within and between groups were evaluated by a one-way analysis of variance (ANOVA) followed by a multi-comparison Dunnett’s test (α = 0.05), ^a,b^ *p* < 0.01, compared with the positive controls (a: nomilin, b: ascorbic acid) in the biological assay, whereas multi-comparison Tukey’s test was applied to evaluate the extracted yuzu seed fractions (*p* < 0.05). ABTS: 2,2′-azino-bis (3-ethylbenzthiazoline-6-sulfonic acid), DPPH: 2,2-Diphenyl-1-picrylhydrazyl, GAE: gallic acid equivalent, IC_50_: amount of extract required for a 50% reduction of free radical scavenging activity, LA: limonoid aglycone, LG: limonoid glucoside, TPC: total polyphenol content.

## Data Availability

The data presented in this study are available in the article and Appendix A.

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
