# Peer review of "Nomilin from Yuzu Seed Has In Vitro Antioxidant Activity and Downregulates Melanogenesis in B16F10 Melanoma Cells through the PKA/CREB Signaling Pathway"

_antioxidants, 2022, doi:10.3390/antiox11091636_

Round 1

Reviewer 1 Report

The manuscript: Nomilin Isolated from Yuzu Seed Shows In Vitro Antioxidant Activity and Downregulates Melanogenesis in B16F10 3 Melanoma Cells via PKA/CREB Signaling Pathway describe some interesting findings. However there are some critical issues that should be resolved before it receives further consideration.

The text is very confusing and poorly writen. A complete revision of the English language used in the manuscript is mandatory.

Page 2 Lines 56-58."Tyrosine is oxidized to DOPA and DOPA quinone by tyrosine?? The authors meant,  tyrosinase?

Page 2 Lines 52 - 54. "Black-brown pigment and metallic protein": These are lay terms. The molecular structure of melanin has to be described in technical terms.

Page 4. Why the activity of gallic acid was used as a control? If gallic acid has a similar structure of nomilin, this has to be shown in the study. Ascorbic acid also shows DPPH radical scavenging activity (Molecules. 2021 Apr 28;26(9):2561. doi: 10.3390/molecules26092561. PMID: 33924795;), why ascorbic acid wasn´t used as a control in the assay?

Page7: There are a lot of criticisms on the use of 2,7-dichloredihydrofluorescein as a probe for detection of ROS production. Detection of Superoxide using the oxidation of Di-hydroethidine and the detection of Hydrogen peroxide using the Amplex Red probe are more appropriated methodologies  

Page 7: This is a very poor description of the westernblot analysis performed by the authors. Authors have to show more details on the analysis performed.

Page 8: Lines 355-356. Free radicals cannot terminate the chain oxidation reaction. They take electrons from reducing substances and become reduced. The reducing substance becomes oxidized.  

Page 9. Lines 392 - 398. This paragraph made this reviewer wonder if these results are repetition of previous findings.

Page 14. Line 577-583. Once more the authors mention previous studies where they found similar findings to the ones described  in the present manuscript.

Page 11 Fig. 3: The Yuzu fruit is rich in Ascorbic acid. Since ascorbic acid is the best inhibitor of mushroom tyrosinase why the authors are using nomilin for the same purpose?  Why ascorbic acid wasn´t used as a comparative control throughout the study?

Page 12. Lines 494-503. This explanation can be found in Biochemistry Text Books. It is absolutely unnecessary to explain what means the Km and Vmax values used in the study to determine the inhibition constant of nomilin  against tyrosinase.

Page 15.  Figure 7. How the morphological changes in B16F10 cells relate to melanin production by the cells when they were incubated with nomilin? Once more, Ascorbic adid should be added as a control since it is a good inhibitor of tyrosinase.

Page 18 Lines 665-667. The authors claim that nomilin´s antioxidant and skin-whitening properties are related to the activation of MEK/ERK and PI3k/Akt signaling pathways. However, they did not show any result to back up this statement.  

Author Response

Answers to Reviewer’s Comments

We thank the reviewers for the feedback on our manuscript and for providing valuable comments that have benefitted the overall manuscript. We have revised the manuscript according to the comments made. Our responses (in red font) to each comment have been provided in the document that follows. Additionally, the revised sections of the manuscript are presented in red font, for your convenience.

Reviewer 1

The manuscript: Nomilin Isolated from Yuzu Seed Shows In Vitro Antioxidant Activity and Downregulates Melanogenesis in B16F10 3 Melanoma Cells via PKA/CREB Signaling Pathway describe some interesting findings. However there are some critical issues that should be resolved before it receives further consideration.

The text is very confusing and poorly written. A complete revision of the English language used in the manuscript is mandatory.

  • The whole manuscript was edited by a professional English Editing Company. From article title to references, we corrected the typos and errors and made the sentence style consistent as well as Figure and Table captions.

Page 2 Lines 56-58."Tyrosine is oxidized to DOPA and DOPA quinone by tyrosine?? The authors meant,  tyrosinase?

  • We changed the last tyrosine to tyrosinase.

Page 2 Lines 52 - 54. "Black-brown pigment and metallic protein": These are lay terms. The molecular structure of melanin has to be described in technical terms.

  • The entire sentence has been revised according to your comment.

Page 4. Why the activity of gallic acid was used as a control? If gallic acid has a similar structure of nomilin, this has to be shown in the study. Ascorbic acid also shows DPPH radical scavenging activity (Molecules. 2021 Apr 28;26(9):2561. doi: 10.3390/molecules26092561. PMID: 33924795;), why ascorbic acid wasn´t used as a control in the assay?

  • It is a typo. We corrected gallic acid to ascorbic acid.

Page7: There are a lot of criticisms on the use of 2,7-dichloredihydrofluorescein as a probe for detection of ROS production. Detection of Superoxide using the oxidation of Di-hydroethidine and the detection of Hydrogen peroxide using the Amplex Red probe are more appropriated methodologies  

  • Thanks for your great comment. In the next study, I will apply the method you suggested.

Page 7: This is a very poor description of the western blot analysis performed by the authors. Authors have to show more details on the analysis performed.

  • We have revised the entire experimental method section.

Page 8: Lines 355-356. Free radicals cannot terminate the chain oxidation reaction. They take electrons from reducing substances and become reduced. The reducing substance becomes oxidized.  

  • The sentence has been corrected according to your comments.

Page 9. Lines 392 - 398. This paragraph made this reviewer wonder if these results are repetition of previous findings.

  • References have been changed.

Page 14. Line 577-583. Once more the authors mention previous studies where they found similar findings to the ones described  in the present manuscript.

  • You are right. But the one study is original experimental article and the other is review paper. In both papers, the positive control used is arbutin.

Page 11 Fig. 3: The Yuzu fruit is rich in Ascorbic acid. Since ascorbic acid is the best inhibitor of mushroom tyrosinase why the authors are using nomilin for the same purpose?  Why ascorbic acid wasn´t used as a comparative control throughout the study?

  • This study aimed to verify the whitening activity of nomilin contained in yuzu seed husk, which had not been revealed in previous studies, and compared the whitening activity of ascorbic acid and nomilin in the tyrosinase inhibition experiment. While, arbutin used for some study because it is one of the best commercial ingredients.

Page 12. Lines 494-503. This explanation can be found in Biochemistry Text Books. It is absolutely unnecessary to explain what means the Km and Vmax values used in the study to determine the inhibition constant of nomilin  against tyrosinase.

  • Thanks for the reviewer's comments, this part has been removed.

Page 15.  Figure 7. How the morphological changes in B16F10 cells relate to melanin production by the cells when they were incubated with nomilin? Once more, Ascorbic adid should be added as a control since it is a good inhibitor of tyrosinase.

  • In this study, various experiments were conducted to verify the whitening activity of nomilin. In the tyrosinase inhibition experiment, ascorbic acid was used as a positive control, and in the melanin contents experiment and western blot experiment, arbutin was used as a positive control. Arbutin is a substance widely used as a positive control in existing whitening activity verification studies.

Page 18 Lines 665-667. The authors claim that nomilin´s antioxidant and skin-whitening properties are related to the activation of MEK/ERK and PI3k/Akt signaling pathways. However, they did not show any result to back up this statement.  

  • Thanks for the critical reviewer comment. There was an error in the sentence, so the sentence was corrected to PKA/CERB cell signaling pathway.

Reviewer 2 Report

The paper submitted by Choi et al. is focused on the evaluation of the bioactivity of natural products - nomilin isolated from Citrus species. The manuscript is well prepared and all experiments are properly panned, reported and performed in my opinion. The paper needs some minor upgrades in order to be published in Antioxidants.

1) the MS spectrum of the isolated compounds including fragmentation in ESI source would be useful for future studies

2) please assign signals for CH3 groups in the NMR reported in section 2.4 - what CH3 groups are not numbered in Fig.2?

2) please include statistical analysis for IC50 values obtained in Table 1 - show statistically significant differences between tested samples

Author Response

Answers to Reviewer’s Comments

We thank the reviewers for the feedback on our manuscript and for providing valuable comments that have benefitted the overall manuscript. We have revised the manuscript according to the comments made. Our responses (in red font) to each comment have been provided in the document that follows. Additionally, the revised sections of the manuscript are presented in red font, for your convenience.

Reviewer 2

The paper submitted by Choi et al. is focused on the evaluation of the bioactivity of natural products - nomilin isolated from Citrus species. The manuscript is well prepared and all experiments are properly panned, reported and performed in my opinion. The paper needs some minor upgrades in order to be published in Antioxidants.

  • the MS spectrum of the isolated compounds including fragmentation in ESI source would be useful for future studies

à Thank you for your suggestion. We will use ESI source for MS study.

2) please assign signals for CH3 groups in the NMR reported in section 2.4 - what CH3 groups are not numbered in Fig.2?

à We edited the NMR data.

3) please include statistical analysis for IC50 values obtained in Table 1 - show statistically significant differences between tested samples

à Statistical methods are added in the experimental method section in 2.15 Statistical Analysis section

Reviewer 3 Report

Dear Authors,

The work is expanded, and many interesting tests have been done however only after some improvements that should be made so that the paper could be considered to accept. Here are my remarks:

1.   Using the terms „limonoid aglycone” and “limonoid glucoside” is a bit confusing. It indicates there is one substance and, in fact, both are fractions.

2.       How was it confirmed that the glycosides contain glucose as a sugar moiety? I could not find it in the manuscript. It should be cleared.

3.       The fractions are not completely defined. Only some ingredients are indicated.

4.       It is also not completely clear why not only nomilin but also limonin contents were assessed. It is a bit confusing because the title and aim of the study refers to nomilin and the work discusses also widely flavonoids contents

5.       There are some gross mistakes as including limonin into flavonoid group (a few times). It

6.       Language style should be improved, some sentences are barely understandable (only for some examples - lines: 54-55, 56 (tyrosine is oxidized by tyrosine?), 289 – not finished sentence, 290 -First, 291, 356-357, 391-392 – the activity expressed as IC50, 398-“citron”? or “yuzu”?, 408-409, 411 (Siegen?) and so on).

7.       The formula in line 209 should be checked.

8.       No statistics have been calculated in anti-oxidant tests. In statistics presented in Figure 6a it is not marked versus which group the comparison was made?

9.       Generally, it looks like every part was prepared by other person/group. The unification in style is missing.

10.   Why was used different substrates for tyrosinase in tyrosine inhibition assay (2.7.1) and enzyme kinetic assay (2.7.2)?

11.   There are some mental shortcuts: e.g. What does it mean that cytotoxicity was less than 100 μg/mL? (line 553). Does it refer to figure 6a or 6b (next line)?

Good luck!

Author Response

Answers to Reviewer’s Comments

We thank the reviewers for the feedback on our manuscript and for providing valuable comments that have benefitted the overall manuscript. We have revised the manuscript according to the comments made. Our responses (in red font) to each comment have been provided in the document that follows. Additionally, the revised sections of the manuscript are presented in red font, for your convenience.

Reviewer 3

The work is expanded, and many interesting tests have been done however only after some improvements that should be made so that the paper could be considered to accept. Here are my remarks:

  1. Using the terms „limonoid aglycone” and “limonoid glucoside” is a bit confusing. It indicates there is one substance and, in fact, both are fractions.
  • In this study, extraction was carried out by dividing yuzu seeds into limonoid glucoside, a hydrophilic component, and limonoid aglycone, a hydrophobic component, and the activity of each fraction was verified. Therefore, the two components are not the same fraction, and our study target material is nomilin contained in the limonoid aglycone fraction.

  1. How was it confirmed that the glycosides contain glucose as a sugar moiety? I could not find it in the manuscript. It should be cleared.
  • We have corrected it to glucoside. That means glucose is a sugar moiety.

  1. The fractions are not completely defined. Only some ingredients are indicated.

à The fractions completely defined in Fig 1.

  1. It is also not completely clear why not only nomilin but also limonin contents were assessed. It is a bit confusing because the title and aim of the study refers to nomilin and the work discusses also widely flavonoids contents.

à In our previous studies, extract content and whitening activity of limonin were reported, but extract content and whitening activity of nomilin were not reported. Therefore, in this study, the content and whitening activity of the existing limonin and nomylin were compared. As a result of this experiment, the content of nomilin was measured to be higher than that of limonin, and the whitening activity was also confirmed to be higher.

  1. There are some gross mistakes as including limonin into flavonoid group (a few times). It

à It is revised as you suggested.

  1. Language style should be improved, some sentences are barely understandable (only for some examples - lines: 54-55, 56 (tyrosine is oxidized by tyrosine?), 289 – not finished sentence, 290 -First, 291, 356-357, 391-392 – the activity expressed as IC50, 398-“citron”? or “yuzu”?, 408-409, 411 (Siegen?) and so on).

à It is revised as you suggested.

  1. The formula in line 209 should be checked.

à It is revised as you suggested.

  1. No statistics have been calculated in anti-oxidant tests. In statistics presented in Figure 6a it is not marked versuswhich group the comparison was made?

à Statistical methods are indicated in the experimental method section (2.15 Statistical Analysis).

  1. Generally, it looks like every part was prepared by other person/group. The unification in style is missing.

à English has been edited for the style unification.

  1. Why was used different substrates for tyrosinase in tyrosine inhibition assay (2.7.1) and enzyme kinetic assay (2.7.2)?

à In general, it is known that the two methods are complementary. The two methods confirmed the inhibition activity in terms of the different substrates.

  1. There are some mental shortcuts: e.g. What does it mean that cytotoxicity was less than 100 μg/mL? (line 553). Does it refer to figure 6a or 6b (next line)?

à In this study, MTT reagent was used to evaluate cytotoxicity, and the cytotoxicity of nomilin did not appear at concentrations below 100 μg/mL.

Round 2

Reviewer 1 Report

Authors showed an improved version of the previous manuscript. This reviewer feels that most of the questions raised in the first round of revision were not properly addressed. Experiments with ascorbic acid that were suggested by this reviewer should be performed.  

Author Response

11 August, 2022

Thank you very much for your valuable comments for our manuscript. I am sure the comments will improve the manuscript considerably. Here, I wrote the answers faithfully according to the reviewer's comments.

Best regards,

Hyun-Jae Shin, Ph.D.

Answers to Reviewer’s Comments

General comments from Editor

(I) Please check that all references are relevant to the contents of the manuscript.

All references have been reviewed and double-checked. Reference number 18 has been deleted. Total number of references are 57.

(II) Any revisions to the manuscript should be marked up using the “Track Changes” function if you are using MS Word/LaTeX, such that any changes can be easily viewed by the editors and reviewers.

The revised sections are marked in red. Especially, the followings were made.

Abstract: Further revisions have been made to the abstract to fit it within the 200-word limit.

Abbreviations: The authors defined all abbreviation throughout the manuscript for clarity.

(III) Please provide a cover letter to explain, point by point, the details of the revisions to the manuscript and your responses to the referees’comments.

I wrote the answer faithfully according to the reviewer's comments.

(IV) If you found it impossible to address certain comments in the review reports, please include an explanation in your rebuttal.

I wrote the answer faithfully according to the reviewer's comments.

(V) The revised version will be sent to the editors and reviewers.

Yes, I will.

Reviewer 1

Q: Authors showed an improved version of the previous manuscript. This reviewer feels that most of the questions raised in the first round of revision were not properly addressed. Experiments with ascorbic acid that were suggested by this reviewer should be performed.

A: Thank you for your valuable comment. Experiments with ascorbic acid were performed and included in the manuscript. As a research team, we have prior experience in conducting antioxidant experiments (DPPH radical scavenging) using ascorbic acid as a positive control in previous studies. In current study, the antioxidant activity (DPPH radical scavenging) of the three extracts of limonoid glucoside and the three extracts of limonoid aglycone extracted from yuzu seeds were compared. In current experiment, the antioxidant activity of the yuzu seed extract and nonillion was compared, and since nomilin is a single compound, it was used as a positive control. However, we fully agrees with the reviewer's opinion, and address the point raised, the DPPH radical experiment was conducted again using ascorbic acid as a positive control (Table 1). As a result of the experiment, the DPPH and ABTS radical scavenging IC50 value of ascorbic acid was found to be 56.97 and 39.95 ug/mL, respectively (supplementary data). Our previous study where we used ascorbic acid as a positive control is provided below for your reference:

Choi, M. H., Jo, H. G., Yang, J. H., Ki, S. H., & Shin, H. J. (2018). Antioxidative and anti-melanogenic activities of bamboo stems (Phyllostachys nigra variety henosis) via PKA/CREB-mediated MITF downregulation in B16F10 melanoma cells. International journal of molecular sciences19(2), 409.

Reviewer 3 Report

I could not accept the paper in this form because the Authors did not understand some of my comments.

1.       A “defined extract” means that all the ingredients of an extract are specified. It was not done in the work.

2.       There is still no statistical significance calculated for the values presented in Table 1. And still, in the Figure 6a it should be marked to which group are compared other groups. It is probably the control group, but it should be marked below the Figure.

3.       According to the 11. comment. I understand what it means, but it is a mental shortcut. And that was only an example as there were a few such expressions in the text. They should not appear in the scientific work because it is not precise. My intention was to suggest changes in the article to make it at higher level.

4.       There is still the expression “limonoid aglycone” and “limonoid glucoside” in the text indicating there is only one compound in each fraction. It is confusing for the reader.

It was only little scientific reflection made and the paper could be accepted only if it is further improved.

Author Response

Dear Reviewer:

Thank you very much for your invaluable comments. It will improve our manuscript considerably.

Best regards,

Hyun-Jae Shin, Ph.D.

Reviewer 3

I could not accept the paper in this form because the Authors did not understand some of my comments.

  1. A “defined extract” means that all the ingredients of an extract are specified. It was not done in the work.

A: Thank you for your insightful comments. In this study, we used eight polyphenols for HPLC analysis: chlorogenic acid, rutin, p-coumaric acid, naringin, hesperidin, luteolin, linoleic acid, and caffeic acid; these eight ingredients are found in large amounts in yuzu seeds, which has been reported in a previous study (shown in supplementary data). Also, in this study, the contents of limonin—found in large amounts in yuzu seeds and well known as a whitening substance—and nomilin—which has not been reported before—were compared and analyzed using HPLC equipment. Both substances were present in the limonoid aglycone fraction, and as a result of the analysis, the content of the two substances was similar. Therefore, based on the HPLC analysis results, the target material, nomilin, was fractionated using MPLC and prep-HPLC equipment, the structure was identified through NMR, and the whitening activity was verified using various assay methods. This study was conducted with reference to the extraction method of reference No. 4. The fraction names were after reference 4.

  1. There is still no statistical significance calculated for the values presented in Table 1. And still, in the Figure 6a it should be marked to which group are compared other groups. It is probably the control group, but it should be marked below the Figure.

A: The text has been revised as per the suggestion provided above.

  1. According to the 11. (Comment 11. There are some mental shortcuts: e.g., What does it mean that cytotoxicity was less than 100 μg/mL? (line 553). Does it refer to figure 6a or 6b (next line)?). I understand what it means, but it is a mental shortcut. And that was only an example as there were a few such expressions in the text. They should not appear in the scientific work because it is not precise. My intention was to suggest changes in the article to make it at higher level.

A: This section was written in English and attached to the main results and discussion section, and references were added to the submitted manuscript.

  1. There is still the expression “limonoid aglycone” and “limonoid glucoside” in the text indicating there is only one compound in each fraction. It is confusing for the reader. It was only little scientific reflection made and the paper could be accepted only if it is further improved.

A: We extracted limonoid aglycone from citron seeds by referring to the previous paper (ref.4), and since it was extracted by the same experimental method, this fraction was confirmed as limonoid aglycone. According to previous research reports, the types of limonoid aglycone extracted from citron seeds include nomilin, obacunone, deacetyl nomilin, and limonin. In this study, the content of the target substances nomilin and limonin, which are known to have the highest concentration, was confirmed.

Ref.4 Minamisawa, M.; Yoshida, S.; Uzawa, A. The functional evaluation of waste yuzu (Citrus junos) seeds. Food Funct. 2014, 5(2), 330-336.